# Oral Health Promotion under the 8020 Campaign in Japan—A Systematic Review

**DOI:** 10.3390/ijerph20031883

**Published:** 2023-01-19

**Authors:** Sachiko Takehara, Raksanan Karawekpanyawong, Hikaru Okubo, Tin Zar Tun, Aulia Ramadhani, Fania Chairunisa, Azusa Tanaka, F. A. Clive Wright, Hiroshi Ogawa

**Affiliations:** 1Division of Preventive Dentistry, Department of Oral Health Science, Faculty of Dentistry & Graduate School of Medical and Dental Sciences, Niigata University, Niigata City 951-8514, Japan; 2Department of Community Dentistry, Faculty of Dentistry, Mahidol University, 10400 Bangkok, Thailand; 3Centre for Education and Research on Ageing, Concord Repatriation General Hospital, Faculty of Medicine and Health, The University of Sydney, Sydney, NSW 2139, Australia; 4Concord Clinical School, Faculty of Medicine & Health, The University of Sydney, Sydney, NSW 2139, Australia

**Keywords:** the 8020 Campaign, oral health, tooth loss, health policy, 8020, Japan

## Abstract

(1) Background: The aim of this study is to review the benefits of the 8020 Campaign since its inception. (2) Methods: We followed the PRISMA guideline and collected information regarding the 8020 Campaign through online database searches. (3) Results: Twenty-five studies met the inclusion criteria and were eligible for analysis. The main outcomes of the 25 included studies were reviewed. The quality evaluation demonstrated a range of studies showing a credible relationship between masticatory function, number of teeth, salivary secretion, frequent dental check-ups, and general health concerns. Due to the risk of bias, publication bias, and indirectness, 22 studies were considered that only had “fair” quality. (4) Conclusions: The 8020 Foundation funded several of the studies, and other research papers noted the 8020 Campaign in their papers, however there were no clear explanations for any direct relationship between their findings and the 8020 Campaign. As a result, evidence for the direct effectiveness and benefits assessment of the 8020 Campaign positive outcomes were compromised by confounding social and economic variables over the 30-year period. To fully elucidate how improvement in Japan’s oral health was directly related to the 8020 Campaign, a more informed and systematic explanation of the campaign’s structure and activities is required.

## 1. Introduction

Major oral diseases and non-communicable diseases (NCDs) are closely associated and shared common risk factors [1,2]. Untreated oral diseases, such as dental caries and periodontal disease, can lead to tooth loss. Oral diseases can also impact overall health conditions [3,4]. Based on the previous research, tooth loss is associated with various poor health conditions [5]. Most older adults experience significant tooth loss; this becomes a specific concern in the Western Pacific Region which has fastest growing aging population. Although Japan has a highest percentage of older adults aged 65 and above in the world, it showed the lowest tooth loss rate [6]. Universal health insurance was introduced in Japan in 1961 and covered most dental treatments [7]. To tackle the high prevalence of oral diseases, the Japanese government initiated the 8020 Campaign in 1989, aiming to create a healthy and long-living society by building public policies, creating supportive environments, developing personal skills, and reorienting health services [8].

The 8020 Campaign is a long-term national oral health promoting strategy [9]. The rationale for the 8020 Campaign was built on the premise that if people can keep at least 20 functioning teeth by 80 years of age, they have a better chance of remaining healthy because of their ability to chew effectively, eat a range of foods, and maintain good nutrition, thereby providing a positive influence on general health and well-being [8]. 

Prior to this, there was a high prevalence of dental caries, gingival and periodontal diseases, and tooth loss among Japanese citizens [8]. The oral health improvement has been monitored through the national health surveys conducted every 6 years since 1957. Results of these national surveys suggest that the oral health of the Japanese population has extensively improved since the introduction of the 8020 Campaign [10]. There have been numerous studies on the effectiveness and consequences of the 8020 Campaign, including the association between the campaign’s efforts and general or oral health [11,12,13,14]. However, the consistency and extent of any direct link between the 8020 Campaign exposure and tooth loss, for example, are unknown. There has been no comprehensive systematic review of the Campaign’s effectiveness. During the 30 years of the 8020 Campaign, Japanese society at large has also experienced changes in social, economic, and public health services as well as nursing care [15]. These changing social determinants and dynamics and the benefit of universal health insurance which enables people to receive dental care with a minimal copayment need to be identified and controlled for in studies evaluating the success of the 8020 Campaign on both oral and general health outcomes. The aim of this study therefore is to critically review the literature pertaining to the major benefits and outcomes of the 8020 Campaign. We hypothesized that the 8020 Campaign had positive impacts on improving oral health and health conditions of the Japanese population over the period of its implementation.

## 2. Materials and Methods

### 2.1. Eligibility Criteria

The inclusion criteria are (1) original primary research articles published in peer-reviewed journals between January 1989 and January 2022 using English and Japanese language; (2) grey literature from reliable sources such as government documents or survey reports that contain a clear description of methods, results, and limitation of the studies; (3) articles addressing oral health-promoting activities under the strategy of the 8020 Campaign and their outcomes across generations and any health conditions; (4) population-based surveys, cross-sectional studies, cohort studies, case-control studies, and interventional studies; and (5) studies with a comparison/control group.

The excluded studies were: (1) outcomes not specifically related to the 8020 Campaign and oral health; (2) grey literature in other forms, such as conference abstracts, posters, theses, dissertations, and study protocols; (3) case reports, animal studies, commentaries, and editorials; (4) outcomes which were solely evaluated through self-reporting assessments; (5) studies lacking complete information; (6) studies in which the full-text articles were inaccessible.

We used the PECO/PICO criteria as follows: the population of interest (P) was Japanese people regardless of their age or conditions; exposures (E) or interventions (I) which were oral health-promoting activities related to the 8020 Campaign or the state of having 20 teeth at the age of 80 and over; there was no restriction regarding the characteristics of comparison groups within each included study (C); outcomes of interest (O) were any changes related to oral health or health conditions, health literacy, and health-related behaviors. The PECO/PICO information was extracted from the included studies. 

### 2.2. Search Terms

The search terms included combinations of the following keywords using Boolean operators: “tooth”, “teeth”, “oral”, “dental”, “mouth” and “8020” for PubMed, Web of Science, and EMBASE (English) literature search, and “8020”, “oral health”, “8020運動 (the 8020 Campaign)”, “口腔 (mouth)” for Google Scholar (Japanese and English), “8020” for Ichushi (Japanese) literature search. Grey literature from Google Scholar and the websites of the 8020 Foundation; the Ministry of Health, Labour, and Welfare; and the Japan Dental Association were also reviewed. Additionally, the reference lists of all included articles were checked for potentially relevant studies.

### 2.3. Search Strategy

According to the PRISMA guidelines, [16] we collected information regarding the 8020 Campaign through online database searches from a Japanese academic search interface (Ichushi), PubMed, Web of Science, Google Scholar and EMBASE. From the 2490 results obtained (by setting keywords, we screened by title and relevance), 501 articles that were published from January 1989 to January 2022 were identified. These databases were selected based on a recommendation for a minimum requirement to guarantee adequate and efficient coverage for literature searches in systematic reviews [17,18]. Details of search criteria and data extraction are mentioned in the Appendix A.

### 2.4. Risk of Bias and Quality Assessment

Risk of bias was assessed using GRADE^25^ and NIH^26^ Quality Assessment Tools and results of the assessment are presented in Appendix A). A summary of quality assessment is presented in Table 1.

### 2.5. Data Extraction

The following data were extracted from the included articles: database source, year of publication, author, study design, number of subjects, population, exposure, intervention (related to the 8020 Campaign, number of teeth, oral health conditions, etc.), control or comparison group, outcome, major findings, and funding sources (The 8020 Foundation or other source). In terms of general health, outcomes included any systemic or general health condition related to number of teeth or quality of life [5]. Data extraction from English articles was independently performed by three reviewers (TZ, AR, and FC), while extraction of data from Japanese articles was carried out by the two Japanese reviewers (HO^*^ and AT). Any ambiguities were resolved through discussion between reviewers or consultation with one of the principal investigators (ST and RK).

### 2.6. Data Synthesis

We did a narrative literature review and generated a summary of our findings. Using the NIH quality assessment tool, we prepared tables summarizing our findings from the articles and evaluated the study’s methodological quality. There was no restriction in effect measures of the outcomes. Further analysis, such as sub-group and sensitivity analysis, could not be conducted. 

## 3. Results

By searching Embase, PubMed, Google scholar, Ichushi (Japanese website), and grey literature, 875 studies resulted, as shown in Figure 1. A total of 25 studies met the inclusion criteria and were eligible for analysis after 827 were initially screened by title and abstract and 135 studies were screened by full text. The excluded studies are explained in detail in Appendix A). The quality of the included articles (n = 25) were tested and shown in the Table 1, and the main outcomes of the included studies, in Table 2.

### 3.1. Quality Assessment Results

Table 1 showed the result of quality evaluation. The quality evaluation demonstrated a range of studies showing a credible relationship between masticatory function, number of teeth, salivary secretion, frequent dental check-ups, and general health concerns. The risk of bias, publication bias, and indirectness of 22 studies meant that assessments of the Program were only of “fair” quality. Three articles reporting on occlusal force and oral flora, provided only a small amount of evidence to be considered reliable [11,12,30]. Out of the 22 studies with a “fair” quality assessment, 22 reported association between the number of teeth with positive oral health, such as masticatory function, occlusal force, salivary flow, and general health, such as physical condition and nutrition. This is consistent with previous studies that analyzed the relationship between number of teeth and general health [5]. However, since the study design for these articles are cross-sectional and did not consider the influence of confounding factors, some of them showed low strength of evidence (18 out of 22) and some (15 out of 22) were included in imprecision criteria. It can also be suspected that publication bias underpinned many articles, such as studies that are not published in reputable, refereed, international journals. 

The poor-quality articles have been designated as having high risk of bias as they did not recruit control and comparison groups from the same population as the targeted population. There was no availability of information about confounding factors in these articles. Moreover, the methodology was not properly described, and structures were not properly designed. With the many gaps in these studies, and the many limitations encountered, it is not possible to draw strong conclusions from these reports on the effectiveness and benefits of the 8020 Campaign. 

### 3.2. Oral Health Promotions for Children (0–14 Years Old)

The lists of oral health promotions programs can be found in Table 2. One study reported the effect of the “Tooth Passport” as an oral health promotion under the 8020 Campaign on dental caries in children [19]. The passport users had less of an increase in caries of their first molars, comparing to the ones who did not use the passport. Beginning to reinforce proper oral hygiene programs in childhood significantly reduced the number of decayed and filled teeth in later life [19].

### 3.3. Oral Health Promotions for Middle-Aged Adults (44–60 Years Old)

Four studies reported the outcomes of 8020 campaign related oral health promotion activities in middle-aged adults. Two studies reported the good impact of keeping more teeth to oral health. Individuals with more than 20 current teeth, have higher saliva secretion [20,21]. In addition, functional tooth evaluation and occlusal contact area were significantly associated with saliva secretion [20]. A study from Takemae et al. showed that better oral condition (the number of teeth, periodontal condition, the ability to chew with one’s teeth or artificial teeth, and the satisfaction of total oral function) were comprehensively related to healthier life style and more frequent participation in social activities [21]. The direction of the association between oral condition and life style cannot be demonstrated as this was a cross-sectional study. The other two studies reported the effect of oral health promotion activities under 8020 Campaign. Morita et al. reported that individuals who attend professional tooth cleaning treatment for 6 years or longer have more teeth present [22], showing that regular professional tooth cleaning contributed to prolonged overall tooth life. A study from the 8020 Promotion Foundation reported that lifestyle and oral health care contributed to the mother’s oral conditions. Compared to the National Data, the prevalence of decayed teeth among participants was similar. However, the prevalence of untreated decayed teeth was lower [23].

### 3.4. Oral Health Promotions for Older Adults (>60 Years Old)

A total of 14 studies reported the positive impact of keeping 20 or more teeth for older people. At the same time, 3 out of 14 studies reported oral health condition as the main outcome together with physical strength such as body mass index, bone mineral density, and grip strength [13,25,26]. The main findings reported that the oral and general health conditions of those with a high number of remaining functional teeth showed significantly more positive health conditions and physical state than those with fewer natural teeth [13,25,26]. In terms of masticatory function, the studies reported that older people with more teeth could masticate a wide variety of foods, [27,34] and had both greater occlusal force and higher bone mineral density [13,25]. The occlusal force of older individuals was not affected by aging if the population had 20 or more natural, functioning, teeth, that is, if they achieved 80:20 [11,30]. Oral function exercises were found to be strongly associated with saliva secretion and swallowing capacity [36]. Moreover, the greater number of retained functional teeth, the lower the microbial flora, resulting in better overall oral health [12]. Studies suggested that better dental and mastication ability had a favorable impact on physical well-being and function [14]. However, income and the number of natural teeth present were associated determinants for the utilization of dental health services. Individuals with poor economic conditions and fewer teeth were less likely to have frequent check-ups [24].

## 4. Discussion

This systematic review evaluated studies related to the activities of the 8020 Campaign since its introduction in 1989, i.e., its impact on oral health conditions, health behaviors, and general health issues. Overall, the 8020 Program was associated with a positive impact on oral diseases, oral health behaviors, and reducing the burden of oral conditions related to general health. The evaluation found “good” quality data on the association between having a complete dentition (28 teeth) and a decreased risk of overall mortality [32], which is consistent with other previous studies [38,39,40,41]. However, this study lacks a degree of external validity because of the selection bias that the study only invited 70-year-old people with 20 or more teeth to participate at baseline. Even though the 8020 Foundation financed some of the included studies and was acknowledged in the publications, there was no clear explanation for the direct link between the findings and the 8020 Campaign. 

The main strength of our systematic review lies in the comprehensiveness of the search. Multiple databases were used to search for relevant studies in two languages, as well as the grey literature. It also had no restriction on age groups, allowing us to look at the impact of the 8020 Campaign on all generations. Quality assessment methods also employed a reliable tool proposed by the National Heart, Lungs, and Blood Institutes Evidence Quality Grading System and the National Institutes of Health Critical Appraisal tool [42]. Furthermore, we followed the guidelines of PRISMA. Moreover, to our knowledge, this study is the first to systematically review “the 8020 Campaign” related publications and objectively assess the campaign’s outcome in English.

Nonetheless, there are several limitations to be considered. First, quality assessment lacked uniformity, as no limitations were imposed on outcomes. In addition, many of the studies included in this review were cross-sectional, reporting associations rather than causal relationships between exposure and outcome. The overall strength of the evidence is not strong enough as the only one cohort study in this review had 10-year study period. As a result, it is impossible to establish for certain if there is a true causal relationship with health benefits occurring within the Japanese population and the 8020 Campaign. Furthermore, most of included articles focused on oral health of older adults, and only two studies examined oral health in the younger generation. Because of this, we cannot conclude anything about younger generations from our study. Aside from that, we were unable to find any comprehensive methods that were utilized to evaluate the impact of the campaign. As a result, the literature reviewed did not allow a high-quality assessment of the 8020 Campaign’s overall direct effectiveness on health improvements.

Despite the above limitations, findings and interpretations of this research have important implications. First, this study showed that the current evidence is insufficient to support promotion of oral health by the 8020 Campaign. It has been reported that the 8020 Campaign has been successful as the results of the Japanese national survey 2016 showed that more than half of the people aged 80 years retained 20 or more teeth. For this reason, the Japanese Society of Oral Health has proposed a new goal as “28 teeth for Lifetime”, which should follow after the 8020 Campaign [43]. However, as indicated from this review, the outcomes of the 8020 Program achievements have not been well evaluated in terms of reliable evidence. The outcomes of the 8020 Campaign need further evaluation by longitudinal and interventional studies. Second, this study has indicated the importance of a future-oriented, lifelong approach [6] to oral health and general health. Although life-course orientation is recognized as important, there was only one study with children as a targeted population found in this systematic review, while the majority of included studies targeted older adult [19]. Early life exposures are important as they can influence an individual’s future health development. As the evidence on life-course approach to oral health is still limited, more studies, especially in early life-stages, need to be conducted [44].

In Japan, the 8020 Campaign may have benefited not only the Japanese people of all generations but also the community at large. Many beneficial effects of oral and general health conditions, health literacy, and health-related behavior have been identified in this review. The studies explained that there was an increase in the oral health program budget since the 8020 Campaign was launched. This too may be a positive outcome associated with the Campaign. Budget and financial aspects were not discussed in our review; however, budgeting and planning are important to support the 8020 Campaign and its evolution. The unique benefits of the 8020 Campaign in terms of improving general health and oral health are difficult to separate from the integration of oral health into the broader health promotion framework in Japan. Both the 8020 Campaign and other health promotion activities across the studied time period have addressed common risk factors such as diet, hygiene, smoking, alcohol use, stress, and trauma [45]. Considering other campaigns in the world, the Pan American Health Organization (PAHO) has been trying to strengthen oral health services through its Oral Health Program, especially for the most vulnerable populations [46]. The FDI World Dental Federation has provided a roadmap regarding possible strategies for a healthy aging society [47]. It is necessary therefore to evaluate the actions and benefits of the 8020 Campaign systematically. As suggested by the WHO Regional Action Plan on Healthy Ageing [6], further research is needed to address the major benefit of the 8020 Campaign and any direct causality of having 20 or more teeth at the age of 80 to oral and general health by using well-designed longitudinal and interventional studies. 

## 5. Conclusions

Given the complex nature of the integration of oral health promotion into a common risk factor approach across the prevention of many chronic health conditions affecting older people, the clear outcomes of the 8020 Campaign could not be determined. To disseminate how Japan promoted oral health through the 8020 Campaign, a more informative and systematic description regarding the structure and activities of the 8020 Campaign needs to be established, and a better designed evaluation of the outcomes of this campaign is urgently warranted. 

## 6. Patents

The current study can be interpreted as a first step in assessing the benefits of the 8020 Campaign. It is crucial to understand the elements of effective public health policy [8], such as financial resources [48], the role of the 8020 Promotion Foundation, and human resource recruiting so that the achievements identified in Japan’s 8020 Campaign toward promoting oral health may be applied in other cultures and countries [49]. Further research could also contribute to a deeper understanding of the significance of hand-to-hand collaboration between the government officials and professionals along with citizens’ compliance in implementing oral health promotion programs in other regions [8]. Although the 8020 activities in children were only partially explained by research [19], it is also still preferable to initiate such a campaign as early as feasible. The key issue, integrated life-course oral health promotion in all generations, showed its effectiveness in preventing tooth loss in older generations. As the FDI World Dental Federation recommended in the roadmap for a healthy aging society [47], governments worldwide have been working to support better oral health for healthy aging. Since Japan is one of the countries with the highest older population, experiences from the 8020 Campaign could provide, an integrated, positive way, to healthy aging.

## Figures and Tables

**Figure 1 ijerph-20-01883-f001:**
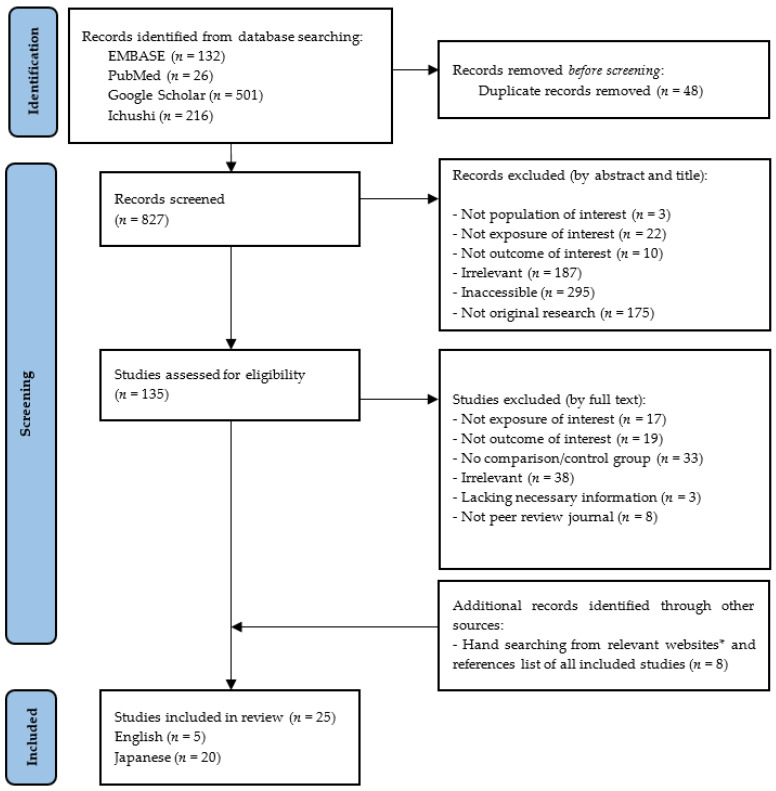
* List of websites: 8020 Promotion Foundation; Japanese Society for Oral Health (JSOH); Ministry of Health, Labour, and Welfare (MHLW).

**Table 1 ijerph-20-01883-t001:** The result of quality assessment.

No	Study	Risk of Bias	Strength of Evidence Grade ^a^	Inconsistency ^b^	Indirectness ^c^	Imprecision ^d^	Publication Bias ^e^	Overall Quality ^f^	
1	Nakane et al. [19]	7	0	0	−1	0	−1	5	Fair
2	Yamamoto et al. [20]	6	−1	0	0	−1	−1	3	Fair
3	Takemae [21]	7	−1	0	−1	−1	−1	3	Fair
4	Morita et al. [22]	9	−1	0	0	−1	1	8	Fair
5	8020 Promotion Foundation [23,24]	6	−1	0	0	−1	−1	3	Fair
6	Motegi et al. [11]	3	−1	1	0	−1	−1	1	Poor
7	Ohazama et al. [12]	3	−1	0	−1	−1	−1	−1	Poor
8	Hashimoto et al. [13]	5	−1	1	0	−1	−1	3	Fair
9	Hashimoto et al. [25]	4	−1	1	−1	−1	1	3	Fair
10	Hirano et al. [26]	7	−1	−1	0	−1	−1	3	Fair
11	Sato et al. [27]	5	−1	1	−1	−1	−1	2	Fair
12	Matsuo et al. [28]	6	−1	1	−1	−1	−1	3	Fair
13	Wakikawa et al. [29]	7	−1	0	−1	−1	−1	3	Fair
14	Takeuchi et al. [30]	5	−1	0	−1	−1	−1	1	Poor
15	Morita et al. [31]	3	−1	1	0	−1	1	3	Fair
16	Iwasaki et al. [32]	11	−1	1	−1	0	1	11	Good
17	8020 Promotion Foundation [24]	8	−1	1	0	0	−1	7	Fair
18	8020 Promotion Foundation [24]	6	−1	0	0	0	−1	4	Fair
19	Kanda et al. [33]	8	0	0	0	0	−1	7	Fair
20	Ansai et al. [34]	6	−1	1	0	0	−1	5	Fair
21	Yamaga et al. [14]	7	−1	0	0	0	1	7	Fair
22	Matsui et al. [35]	6	−1	0	−1	−1	−1	2	Fair
23	Ibayashi et al. [36]	5	0	0	−1	−1	−1	2	Fair
24a	Shikura et al. (1) [37]	6	−1	0	0	0	−1	4	Fair
24b	Shikura et al. (2) [37]	5	0	0	0	−1	−1	3	Fair
25	8020 Promotion Foundation [24]	7	−1	0	0	−1	−1	4	Fair

^a^ (−1 = low,0 = moderate, 1 = high), ^b^ (1 = consistent with other included studies, 0 = NA (different outcome with other included studies, −1 = inconsistent (different findings with other included studies), ^c^ (0 = direct, −1 = indirect), ^d^ (−1 = imprecision, 0 = precision), ^e^ (1 = undetected, −1 = suspected), ^f^ (Good = 16 to 10, Fair = 9 to 2, Poor = 1 to (−5)).

**Table 2 ijerph-20-01883-t002:** The outcomes of the 8020 Campaign from the included studies.

No	Author	Study Design	Population	Exposure/Intervention	Control/Comparison	Outcome		Key Findings	Funding/ Language
						Oral Health	General Health		
**Children Group (0–14 years old)**									
1	Nakane et al., 2013 [19]	Cohort study	198 (M-96, F-102) children who used a tooth passport from 11 elementary schools (Approximately 7 years old at the baseline) ^(a)^	Received oral health-promoting activities related to the 8020 Campaign (Tooth passport)	406 (M-205, F-201) children who did not use a tooth passport from 11 elementary schools (Approximately 12 years old) ^(a)^	The mean number of decayed and filled teeth of study and control groups was evaluated.	-	“Tooth passport” users had a lower increase in caries of their first molars. The mean number of decayed and filled teeth in the control group is significantly higher (*p* < 0.05) than in the tooth passport group for both sexes.	No funding/Japanese
**Middle Age Adults Group (44–60 years old)**									
2	Yamamoto et al., 2011 [20]	Cross-sectional study	34 ^(b)^ adults with 20 or more teeth present and visited the dental clinic in June 2009 (with an average of 55.4 ± 13.7 years and an average of 25.3 ± 2.7 remaining teeth)	Having 20 or more natural teeth present	20 adults with less than 20 teeth present, and visited dental clinic in June 2009 (with an average of 71.9 ± 7.4 years and an average of 9.9 ± 5.8 remaining teeth) ^(b)^	Salivary secretion of study and control groups was evaluated.	-	Saliva secretion was significantly higher (*p* < 0.01) in those with more than 20 current teeth, and functional tooth evaluation value and occlusal contact area were significantly associated with saliva secretion.	No funding/Japanese
3	Takemae, 1996 [21]	Cross-sectional study	33 (M-11, F-32) residents of Tokyo with no treatment need ^(a)^	Maintaining good oral conditions with no treatment need.	110 (M-50, F-82) residents of Tokyo with treatment need (prosthodontic or periodontal treatment) ^(a)^	-	The following were evaluated in study and control groups: Physical fitness: - BMI - Blood pressure - Blood chemistry Social activities, Health Practice index (Modified Breslow’s Health Practices Index)	Better oral condition (the number of teeth, periodontal condition, the ability to chew with one ’s teeth or the artificial teeth, and the satisfaction of total oral function) were comprehensively related to healthier life style and more frequent participation in social activities.	No funding/Japanese
4	Morita et al., 1995 [22]	Cohort study	407 (M-158, F-249) people participated in a 6-year follow-up study (with an average of 47.7 years and an average of 25.5 ± 4.9 remaining teeth)	Receiving regular professional tooth cleaning for 6 years	Average values from a national survey (Survey of Dental Diseases in 1981 and 1987) ^(a) (b) (c)^	Tooth loss for 6 years in study and control groups was evaluated.	-	More teeth were present compared to those in the corresponding age groups. Regular professional tooth cleaning contributed to prolong overall tooth life.	No funding/Japanese
5	8020 Promotion Foundation, 2007 [23,24]	Cross-sectional study	Mothers who visited Infant Dental Check-up in Niigata, Kanagawa, Aichi and Nagasaki prefectures (average age of 31.4 ± 4.5 years) 2786 mothers: received oral examination. 3301 mothers: answered questionnaire	Experienced school-based fluoride mouth rinsing program	Average values from a national survey (Survey of Dental Diseases in 2005) ^(a) (b) (c)^	Oral condition including decayed and missing teeth were evaluated		The prevalence of decayed teeth among participants was similar to national data. However, the prevalence of untreated decayed teeth was lower than national data. Among the 4 prefectures, the average number of missing teeth was lowest in the Niigata area, where school-based fluoride mouth rinsing programs has been conducted.	8020 Promotion Foundation ^¶^ /Japanese
**Older People Group (>60 years old)**									
6	Motegi et al., 2009 [11]	Cross-sectional study	46 (M-22, F-24) older people college students (with an average of 66.9 years and an average of 25.5 remaining teeth)	Number of teeth was evaluated as exposure	52 (M-28, F-24) 8020 achievers (with average age 84.3 years)	Occlusal force of study and control groups was evaluated.	-	The average occlusal force (942.9 ± 440.1 N) was not significantly different between two groups and it was not affected by aging if many teeth are present	No funding/English
7	Ohazama et al., 2006 [12]	Cross-sectional study	22 (M-6, F-16) independent 8020 achievers (with an average age of 81.3 years and an average of 24.7 remaining teeth)	Having 20 or more natural teeth present at 80 years of age	38 (M-10 and F-28) older people living in a nursing home (with average age 81.3 years and average 4.2 remaining teeth)	Number of oral flora (from saliva sample) of study and control groups was evaluated.	-	The average number of oral flora (*Staphylococcus aureus* and *Candida albicans*) were lower in independent 8020-achievers (who have a greater number of teeth), indicating that 8020 achievers have better oral health.	No funding/ English
8	Hashimoto et al., 2006 [13]	Cross-sectional study	217 (M-126, F-91) 8020-achievers (with an average of 81 years and at least 20 teeth at the age of 80 years or more)	Having 20 or more natural teeth present at 80 years of age	104 (M-54, F-50) 8020 non-achievers (with average age 81.8 years (male) and 81.4 years (female))	The following were evaluated in study and control groups: Periodontal condition, Masticatory ability, Saliva flow rate, Salivation buffer, Occlusal force	The following were evaluated in study and control groups: Body mass index, Bone mineral density (BMD), Grip strength, Balance test	Masticatory ability in 8020-achievers was higher than non-achievers. Grip strength was higher in 8020-achievers than non-achievers. Female 8020-achievers had higher BMD than female non-achievers. Similarly, male 8020-achievers had higher score in balance-test than male non-achievers.	No funding/English
9	Hashimoto et al., 2006 [25]	Cross-sectional study	123 (F) 8020 achievers (with an average of 81 years and an average of 24.7 remaining teeth)	Having 20 or more natural teeth present at 80 years of age	80 (F) from institutionalized older people (with an average of 82.4 years old and average 6.3 remaining teeth)	The following were evaluated in study and control groups: Periodontal condition, Masticatory ability, Saliva flow rate, Salivation buffer, Occlusal force	The following were evaluated in study and control groups: Body Mass Index, Bone mineral density (BMD), Grip strength	There is a significant difference between the oral and general health of 8020 achievers and non-achievers. BMD, grip strength, saliva flow rate, and occlusal force were higher in 8020-achievers.	No funding/English
10	Hirano et al., 1993 [26]	Cross-sectional study	405 (M-183, F-222) residents of Tokyo (aged 65 to 84)	Number of teeth was evaluated as exposure	No control	Masticatory function was evaluated	Physical fitness, Bone mineral density (BMD), Grip strength, Balance function were evaluated.	With increasing age, the number of natural teeth (*p* < 0.01) and the masticatory function significantly declined, but there was no significant correlation between age and the number of functioning teeth. The more natural teeth there were, the better the masticatory ability was (*p* < 0.01), but there was no significant correlation between the number of functioning teeth and the masticatory ability.	Tokyo Metropolitan Institute of Gerontology ^¶^/Japanese
11	Sato et al., 2007 [27]	Cross-sectional study	8020-group: 36 (M-21, F-15) older adults who received the 8020 awards in 2006 (with an average of 84 ± 4 years and an average of 24 ± 2 remaining teeth)	Having 20 or more natural teeth present at 80 years of age	8000-group: 35 (M-20, F-15) older adults who received full denture treatment at a dental clinic (with an average of 83 ± 5 years and an average of 0 ± 0 remaining teeth)	The following were evaluated in study and control groups: Swallowing function, Oral dryness, Occlusal force, Chewing function, Oral problems	The following were evaluated in study and control groups: Medico-social conditions Quality of life (QOL)	Although their occlusal force and chewing function were slightly low for the 8000-group (*p* < 0.01), appropriate prosthetic treatment may prevent the decrease of swallowing, medico-social conditions, oral problems, and QOL. For the 8020-group, some people reported problems of “food trapped between teeth” (*p* < 0.05), indigestion (*p* < 0.05), and temporomandibular joint disorder, which might cause the decrease of QOL.	Ministry of Education, Culture, Sports, Science, and Technology 2005–2006 Scientific Research Fund Subsidy Base ^¶^ /Japanese
12	Matsuo et al., 2016 [28]	Cross-sectional study	164 patients who were admitted to the hospital from October 2015 to February 2016, with normal or well-nutrition (aged 75 year over) ^(b)^	Number of teeth and following conditions were evaluated as exposure: Oral wetness, Number of oral bacteria, Tongue pressure, Masticatory ability, Chewing ability, Tongue dexterity and velocity	110 patients who were admitted to the hospital from October 2015 to February 2016, with malnutrition ^(b)^	-	The following were evaluated in study and control groups: Nutrition status Physical fitness - Grip strength - Pinch force Quality of life (QOL), Activities of daily living (ADL)	Most of the oral functions were lower in the malnutrition group, who were also affected by aging. The measures for QOL and ADL were also significantly lower in the malnutrition group.	Japanese Society of Gerodontology Grant-in-Aid for Scientific Research and Japan Society for the Promotion of Science Grant-in-Aid for Scientific Research ^¶^ /Japanese
13	Wakikawa et al., 2013 [29]	Cross-sectional study	103 (M-61, F-42) hemodialysis patients (average of 69.3 ± 10.8 years and an average of 15.2 ± 0.1 remaining teeth)	Number of teeth was evaluated as exposure	103 (M-48, F-55) non-dialysis patients with an average of 67.2 ± 11.8 years and an average of 21.2 ± 7.6 remaining teeth).	The following were evaluated in study and control groups: Caries, Periodontitis	The following was evaluated in study and control groups: Nutrition status	The prevalence of periodontal disease was high in hemodialysis patients with diabetes mellitus. Many hemodialysis patients had periodontal pockets deeper than 4 mm. Nutrition status evaluated by albumin level was significantly lower in people with 9 or fewer teeth in dialysis patients.	No funding/Japanese
14	Takeuchi et al., 2005 [30]	Cross-sectional study	52 (M-28, F-24) 8020 achievers ^(a)^	Having 20 or more natural teeth present at 80 years of age	28 persons with normal occlusion whose ages were in their 20 s. ^(b)^	The following were evaluated in study and control groups: Occlusal force, Occlusal area, Occlusal pressure	-	The occlusal force of 8020- achievers was not significantly different from that of normal occlusal persons in their twenties, but the occlusal force was lower in 8020-achievers with 23 teeth or less.	No funding/Japanese
15	Morita et al., 1996 [31]	Cross-sectional study	54 older people with more than 20 teeth (8020-achievers) (average of 82.6 ± 2.8 years and an average of 23.9 ± 2.8 remaining teeth) ^(b)^	Having 20 or more natural teeth present at 80 years of age	51 older people with 19 or less teeth (average of 82.7 ± 3.2 years and an average of 2.9 ± 4.9 remaining teeth) ^(b)^	-	The following were evaluated in study and control groups: Nutrition status Variety of food intake	The older people who had 20 or more natural teeth tended to have a low intake of energy and wide food variety. For example, low intake of energy and low intake of carbohydrates were observed in the 8020-group compared with the controls. The 8020-achievers ate many kinds of food.	No funding/Japanese
16	Iwasaki et al., 2019 [32]	Cohort study	105 (M-62, F-43) adults aged 70 years who had 20 or more teeth at baseline and maintained 28 teeth at 10 years follow-up	Having 28 natural teeth present at 80 years of age	Adults aged 70 years who had 20 or more teeth at baseline and had 27 or less teeth at 10-year follow up. 103 (M-50, F-53): average 2 tooth loss at 10 years follow up, 91 (M- 42, F- 49): average 4.9 tooth loss at for 10 years follow-up	-	The following was evaluated in study and control groups: Life expectancy	The 28-tooth maintained group had a significantly lower risk of total mortality compared to the tooth loss group (adjusted hazard ratio = 0.50, 95% confidence interval = 0.28–0.89).	No funding/Japanese
17	8020 Promotion Foundation, 2020 [24]	Cohort study	5607 people with more than 20 teeth (20 years or older) ^(b)^	Having 20 or more natural teeth present	904 people with 19 or less teeth (20 years or older) ^(b)^	-	The following were evaluated in study and control groups: Non-communicative disease (NCDs), Diabetes mellitus (DM), Stroke, Cardiovascular disease (CVD), Cancer, Hypertension (HT), Hyperlipidemia.	From the baseline data, there is a trend observed that incidence of DM, stroke, CVD, cancer, HT, hyperlipidemia decreased if more teeth were present.	8020 Promotion Foundation ^¶^/Japanese
18	8020 Promotion Foundation, 2020 [24]	Cross-sectional study	754 (M-294, F-460) Japanese population (20–79 years old)	Receiving regular dental check-ups	1407 (M-714, F-693) without regular dental check-ups (20–79 years old)	Visiting dental clinics for regular dental check-ups was evaluated in study and control groups	-	The utilization of dental clinics for regular check-ups was related to factors such as income and number of teeth. People with fewer teeth and difficult economic conditions were less likely to have regular check-ups.	8020 Promotion Foundation ^¶^/Japanese
19	Kanda et al., 2008 [33]	Cohort study	39,861 (M-17,660, F-22,201) people who received medical and dental treatments in 2002 (with an average age of 75.9 ± 5.0 years and an average of 14.4 ± 8.9 remaining teeth)	Changes in the number of teeth over 3 years	Followed up 3 years. 29,861 (M-13,048, F-29,861) same people who received medical and dental treatments in 2005 (with an average age of 76.4 ± 5.0 years and an average of 15.4 ± 8.8 remaining teeth)	-	The following was evaluated in study and control groups: Annual medical expenditure (2002–2005)	Annual medical expenditure is higher with age, in males, and for those who have fewer teeth, all statistically significant (*p* < 0.001).	8020 Promotion Foundation ^¶^/Japanese
20	Ansai et al., 2000 [34]	Cross-sectional study	116 (M-64, F-52) 80-year-old people (born in 1917)	Exercise function (hand grip strength, one-leg standing time, leg extensor power, stepping rate), Bone mineral density	707 (M-245, F-462) 8020 non-achiever	The following were evaluated in study and control groups: Number of teeth, Masticatory efficiency, Number of chewable foods	-	Significant relationships between the number of teeth and the belief that “we should take care of our own oral hygiene and oral health” (*p* < 0.05) and masticatory efficiency and the number of teeth (≧20) (*p* < 0.01) were found. Elderly people with a high exercise function possessed a higher number of teeth (*p* < 0.01), could masticate a greater number of foods (*p* < 0.05), and had a higher bone density (*p* < 0.01).	Ministry of Health, Labour, and Welfare ^¶^/Japanese
21	Yamaga et al., 2002 [14]	Cross-sectional study	591 (M-302, F-289) older people (aged 70 years)	Dental occlusal condition: Presence of natural tooth contacts between maxilla and mandible in the bilateral premolar and molar regions	158 (M-71, F-87) older people (aged 80 years)	-	The following were evaluated in study and control groups: Physical fitness: - Grip strength - Lex extensor power - Stepping rate - One leg standing time with eyes open	Dental occlusal condition is associated with lower extremity dynamic strength, agility, and balance function in older adults (leg extensor power (R2 = 0.627, *p* < 0.05), stepping rate (R2 = 0.159, *p* < 0.05), and one-leg standing time with eyes open (R2 = 0.179, *p* < 0.05) showed significant correlations with the Eichner index).	Grant-in-Aid for Research on Health Services from the Ministry of Health and Welfare of Japan. ¶/English
22	Matsui et al., 1995 [35]	Cross-sectional study	33(M-7, F-26) older people who live in a nursing home, are independent, and do not need long-term care (aged 76 to 91)	Oral conditions: Chewing ability, Quantity of food intake, Wearing dentures and their conditions	34(M-8, F-26) older people who live in a nursing home, are independent, and do not need long-term care (aged 63 to 75)	-	The following were evaluated in study and control groups: Self-reported anxiety evaluated by STAI-S, STA-T.	Older women who reported good chewing ability and no problem with dentures had less anxiety.	No funding/Japanese
23	Ibayashi et al., 2006 [36]	Before-After study	14 (M-3, F-11) program participants (69–85 years old) who remained until the last day on remote islands	Received oral hygiene guidance, Oral function exercise	Baseline conditions of 22 (M-4, F-18) participants (Mean age: 77.29 ± 4.65, 69 to 85)	The following were evaluated in study and control groups: Swallowing function, Stimulating saliva flow rate, Saliva buffering capacity, S.M. bacterial count, Occlusal force, Number of natural teeth, Number of dentures	-	Oral function exercise for 3 months was effective to improve swallowing function, increase saliva secretion.	2005 Grant from Sompo Japan Research Institute Co., Ltd. ^§^/Japanese
24	Shikura et al., 2020 [37]	(1) Cross-sectional study	(1) 378 (M-217, F-159, Unknown-2) people who worked for tertiary industry such as retailing, wholesales, financing, real estate (with age 18 to 84)	(1) Receiving regular dental check-ups	(1) 269 (M-169, F-99, Unknown-1) people without regular dental check-ups ^(a)^	(1) Regular dental check-ups was evaluated in study and control groups		(1) Receiving regular dental check-ups was associated with no decayed teeth (OR = 2.24), oral health literacy (OR = 3.62), using interdental brushes (OR = 2.41), and using floss (OR = 2.09).	No funding/Japanese
		(2) Intervention study	(2) 11 people with tooth brushing instruction ^(a) (b)^	(2) Received tooth brushing instruction once in 6 months for 2 years	(2) 10 people with no tooth brushing instruction ^(a) (b)^	(2) Oral hygiene condition evaluated by plaque index was evaluated in study and control groups:	-	(2) Oral hygiene condition evaluated by plaque index improved.	
25	8020 Promotion Foundation, 2020 [24]	Cohort study	3194 (M-1001, F-2193) patients who visited dental clinics solely for check-ups for 5 years from 2014 to 2019 (20 years or older)	Received regular dental check-ups for 5 years	6371 (M-1307, F-2264) patients followed up from 2014 to 2019, and visited dental clinics for other reasons (20 years or older)		The following were evaluated in study and control groups: Self-reported health condition, Incidence of NCDs	Visiting dental clinics annually for regular check-ups was associated with self-reported health conditions. People who visited dental clinics for regular check-ups evaluated their health conditions higher than those who did not.	8020 Promotion Foundation ^¶^/Japanese

^¶^ Public sector ^§^ Private sector. ^(a)^ No detailed information about age was provided in the article ^(b)^ No detailed information about gender was provided in the article. ^(c)^ No detailed information on number of persons was provided in the article.

## Data Availability

The data supporting this study’s findings are available from the corresponding author upon reasonable request.

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
