# Peer review of "Oral Health Promotion under the 8020 Campaign in Japan—A Systematic Review"

_ijerph, 2023, doi:10.3390/ijerph20031883_

Round 1

Reviewer 1 Report

Dear Editor,

First, I would like to congratulate the authors for their complex work in writing the manuscript entitled „Oral Health Promotion under the 8020 Campaign in Japan 2

– A Systematic Review –„. The research is well structured, and the results are correctly presented. However, I have some suggestions regarding the manuscript:

-       In abstract the conclusion section resembles more like the discussion. Please shorten this section and extend the results section, including more data. 

-       In the end of the manuscript, the conclusion section is too long. I suggest shortening and including any supplementary comments in the discussion section. 

Author Response

Thank you for giving us the opportunity to submit a revised draft of our manuscript. Our detailed response to the reviewer’s comments is attached below.  We would also like to thank the reviewer for their suggestions which we believe have improved our paper.

Comment 1

In abstract the conclusion section resembles more like the discussion. Please shorten this section and extend the results section, including more data. 

Response:

In the abstract, the conclusion has been shortened and content of the results has been added.

Comment 2

 In the end of the manuscript, the conclusion section is too long. I suggest shortening and including any supplementary comments in the discussion section. 

Response:

As the reviewer commented the conclusion of the main text has been shortened.

Reviewer 2 Report

Dear authors. 

The study is an interesting work that exposes how the scientific literature and other types of publications have reviewed the work carried out in Japan's 80-20 campaign.

This work is required to later escalate to a more extensive study that includes the results obtained in the campaign, with a strong scientific component.

The main complication for this review was that most of the articles were written in Japanese, so it was especially complex to analyze the relevance of what was described in the text.

It is an adequate study for the journal's objectives, but I suggest to the authors some considerations that could improve the manuscript.

INTRODUCTION:

In the first line, please define the acronym NCDs.

The introduction story provides essential information to understand the objectives of the 8020 campaign. However, it is recommended that the writing be improved by eliminating the repetition of ideas that make the writing unnecessarily long. For example, in the second paragraph, the description of the 8020 campaign rationale makes the general objective described in lines 45 and 46 redundant.

It is suggested to describe a hypothesis for this work.

METHODOLOGY:

In the description of the PICO criteria, I have two doubts:

First, why is “the state of having 20 teeth at the age of 80 and over” considered an Intervention? This aspect seems to be an Outcome of what is expected of the 8020 campaign. 

Second, the criteria used for the Comparison groups are unclear. Does it mean that all the Comparisons made in each study selected for analysis were included? Please clarify to maintain consistency between the methodology and the results described in table 2.

In the search strategy, was only “8020 campaigning” used as the keyword, or were other keywords also used? Please describe the keywords used in the search.

RESULTS

In table 2 of the results, the concept "number of teeth" is considered as exposure (E) or intervention (I). It is necessary to express this better and make it more understandable to the reader what is being intervened or exposed. If I am correct, this part describes the intervention that is being carried out to maintain a certain number of teeth in the mouth (20 teeth at the age of 80 and over). Perhaps the authors should describe the intervention as concrete action, such as “oral health approach strategies to maintain a number of teeth of 20 in adults over 80 years of age”.

I also suggest making the outcomes (O) more explicit so that the reader can better correlate the exposure/intervention, control/comparator, and outcomes.

It is mentioned that patients who attend professional tooth cleanings retain more teeth. Is it the only treatment these patients received? Please clarify.

I think the correlation between oral health and "lifestyle" is debatable. Since the focus of the analysis is from oral health to lifestyle, it is suggested to discuss the other direction: how lifestyle can affect oral health.

I suggest referring to the bias involved in reviewing studies carried out by those who promote health policies and the 8020 campaign since these reports may have a different objective than published scientific studies.

DISCUSSION

The manuscript reveals a problem in published studies regarding the 80-20 campaign.

The discussion mentions a specific result concerning the 8020 campaign: the Japanese national survey 2016. The study revised the literature to verify if there is evidence that supports or questions the expected effect of the 8020 campaign. According to the results, the perception of the outcome described in the national survey cannot be confirmed or refuted since the information is mainly of fair quality, and more studies must be conducted. As mentioned, the main result observed is the confirmation that there is insufficient scientific information to evaluate the 8020 campaign. Therefore, an adequate correlation between the scientific evidence and the “perception” of the results cannot be made. Due to the above, it is probably inappropriate to refer to a “gap.” If the hypothesis is based on the difference between “perception” and scientific evidence, this must be described. Consequently, I see two lines of discussion in the study: A criticism of the studies that have been carried out regarding the 8020 campaign and the lack of good quality studies by the promoters of the 8020 campaign.

On the other hand, it seems logical that a campaign aimed at maintaining 20 teeth in 80-year-olds generates studies mainly focused on older adults, so the authors should explain or argue more widely why it is necessary to carry out studies in younger people.

Finally, the authors should make an effort to discuss these results considering other campaigns in the world, such as the one carried out by the World Health Organization with the Pan American Health Organization; this is intended to involve the results of 8020 with global oral health campaigns.

Author Response

Thank you for giving us the opportunity to submit a revised draft of our manuscript. Our detailed response to the reviewer's comments is attached below.  We would also like to thank the reviewer for their suggestions which we believe have improved our paper.

INTRODUCTION

Comment 1

In the first line, please define the acronym NCDs.

Action/ Change to Manuscript:

Definition of NCDs has been added (line 32).

Comment 2

It is recommended that the writing be improved by eliminating the repetition of ideas that make the writing unnecessarily long. For example, in the second paragraph, the description of the 8020 campaign rationale makes the general objective described in lines 45 and 46 redundant.

Response:

We carefully checked the manuscript and eliminated redundancy as much as possible.

Comment 3

It is suggested to describe a hypothesis for this work.

Response:

We have added a research hypothesis along with a hypothesis in the last paragraph of the Introduction.

Added sentence: “We hypothesized that the 8020 Campaign had positive impacts on improving oral health and health conditions of the Japanese population over the period of its implementation.”

METHODOLOGY

Comment 4

1)     In the description of the PICO criteria, I have two doubts:

First, why is “the state of having 20 teeth at the age of 80 and over” considered an Intervention? This aspect seems to be an Outcome of what is expected of the 8020 campaign. 

2)     Second, the criteria used for the Comparison groups are unclear. Does it mean that all the Comparisons made in each study selected for analysis were included? Please clarify to maintain consistency between the methodology and the results described in table 2.

3)     In the search strategy, was only “8020 campaigning” used as the keyword, or were other keywords also used? Please describe the keywords used in the search.

Response:

1)     Regarding the first comment, retaining 20 or more teeth at the age of 80 is the objective of the 8020 Campaign as the reviewer mentioned. However, the effect of retaining more natural teeth through the 8020 Campaign needed to be established as a rationale for promoting the campaign. For this reason, we defined our intervention as retaining more natural teeth, and the outcome as any changes related to oral health or health conditions.

2)     As for the second comment, comparison groups were selected based on study designs. In the case of intervention studies, comparison groups were defined in their studies. The text was revised as it follows: “there was no restriction regarding the characteristics of comparison groups, and information of comparison groups within each included study” (line 89).

3)     We added search keywords in the materials and methods section. Our apology that the search terms were listed as the supplemental material (line 92-100).

RESULTS

Comment 5

In table 2 of the results, the concept "number of teeth" is considered as exposure (E) or intervention (I). It is necessary to express this better and make it more understandable to the reader what is being intervened or exposed. If I am correct, this part describes the intervention that is being carried out to maintain a certain number of teeth in the mouth (20 teeth at the age of 80 and over). Perhaps the authors should describe the intervention as concrete action, such as “oral health approach strategies to maintain a number of teeth of 20 in adults over 80 years of age”.

Response:

Thank you very much for the comment. We revised the table 2 to make exposures/interventions more understandable (line 260).

Comment 2

I also suggest making the outcomes (O) more explicit so that the reader can better correlate the exposure/intervention, control/comparator, and outcomes.

It is mentioned that patients who attend professional tooth cleanings retain more teeth. Is it the only treatment these patients received? Please clarify.

Response:

Thank you very much for the comment. We revised the table 2 in order to clarify the outcomes accordingly (line 260).

Comment 3

I think the correlation between oral health and "lifestyle" is debatable. Since the focus of the analysis is from oral health to lifestyle, it is suggested to discuss the other direction: how lifestyle can affect oral health.

Response:

Thank you very much for the comment. As you pointed out, we don’t know the direction of the association as this is the cross-sectional study. We added this point in the result section 3.3 (line 230-231).

Comment 4

I suggest referring to the bias involved in reviewing studies carried out by those who promote health policies and the 8020 campaign since these reports may have a different objective than published scientific studies.

Response:

Thank you very much for the comment.  I agree that if studies were carried out by those related with the 8020 campaign or funded by any organizations related with the 8020 Campaign, this should be considered as a bias. We considered this bias in our analysis using the risk of bias tool (supplemental material, page 1)

DISCUSSION

Comment 5

The manuscript reveals a problem in published studies regarding the 80-20 campaign.

The discussion mentions a specific result concerning the 8020 campaign: the Japanese national survey 2016. The study revised the literature to verify if there is evidence that supports or questions the expected effect of the 8020 campaign. According to the results, the perception of the outcome described in the national survey cannot be confirmed or refuted since the information is mainly of fair quality, and more studies must be conducted. As mentioned, the main result observed is the confirmation that there is insufficient scientific information to evaluate the 8020 campaign. Therefore, an adequate correlation between the scientific evidence and the “perception” of the results cannot be made. Due to the above, it is probably inappropriate to refer to a “gap.”

If the hypothesis is based on the difference between “perception” and scientific evidence, this must be described.

Response:

Thank you very much for the comment. We deleted the word “gap“ from the text, and revised the sentence as “First, this study disclosed the current evidence is insufficient to support promotion of oral health by the 8020 Campaign“ (line 300-301).

Comment 6

Consequently, I see two lines of discussion in the study: A criticism of the studies that have been carried out regarding the 8020 campaign and the lack of good quality studies by the promoters of the 8020 campaign.

On the other hand, it seems logical that a campaign aimed at maintaining 20 teeth in 80-year-olds generates studies mainly focused on older adults, so the authors should explain or argue more widely why it is necessary to carry out studies in younger people.

Response:

We also added a reason of the necessity to carry out studied in younger generation as follows: “Early life exposures are important as they can influence an individual’s future health development. As the evidence on life-course approach to oral health is still limited, more studies among younger generation need to be conducted“ (line 312-315).

Comment 7

Finally, the authors should make an effort to discuss these results considering other campaigns in the world, such as the one carried out by the World Health Organization with the Pan American Health Organization; this is intended to involve the results of 8020 with global oral health campaigns.

Response:

Thank you very much for the information. We added the discussion about other campaign carried out by WHO with Pan American Health Organization in the last paragraph of the discussion section.

Round 2

Reviewer 2 Report

Dear authors,

Thank you for considering the suggestions made.

Sincerely,

The reviewer.